# Is ChatGPT the ultimate Data Augmentation Algorithm?

**Frédéric Piedboeuf**
RALI, Diro
Université de Montréal
`frederic.piedboeuf@umontreal.ca`

**Philippe Langlais**
RALI, Diro
Université de Montréal
`felipe@iro.umontreal.ca`

## Abstract

In the aftermath of GPT-3.5, commonly known as ChatGPT, research has attempted to assess its capacity for lowering annotation cost, either by doing zero-shot learning, generating new data, or replacing human annotators. Some studies have also investigated its use for data augmentation (DA), but only in limited contexts, which still leaves the question of how ChatGPT performs compared to state-of-the-art algorithms. In this paper, we use ChatGPT to create new data both with paraphrasing and with zero-shot generation, and compare it to seven other algorithms. We show that while ChatGPT performs exceptionally well on some datasets, it overall does not perform better than the other algorithms, yet demands a much larger implication from the practitioner due to ChatGPT often refusing to answer due to sensitive content in the datasets.

## 1 Introduction

Textual data augmentation (DA) is a rich and complicated field, with the goal of finding ways to generate the most informative artificial examples to add to a training set without additional labelling cost. Many techniques have been developed and thoroughly tested in order to create informative artificial data. Recently, the new model ChatGPT has put into question much of what was known for dataset construction, with researchers even wondering if its capacities meant the end of human labelling (Kuzman et al., 2023).

While extensive studies have been done on the use of ChatGPT for many Natural Language Understanding tasks, its use for DA has been surprisingly little investigated yet. To our knowledge, only two papers have looked into this, namely (Møller et al., 2023) and (Dai et al., 2023). Both however explore limited settings, which makes it still unclear whether ChatGPT is a good tool for data augmentation or not. In particular, Møller et al. (2023) study the performance of ChatGPT and GPT-4 on classification tasks of medium size (500 examples), comparing zero shot data generation and few-shot+data augmentation to crowdsourcing annotation, but not to other DA techniques. Dai et al. (2023) compare the use of DA with ChatGPT for large datasets (several thousand examples), focussing on bio-medical data. Their algorithm however does not isolate ChatGPT for data augmentation, but instead combines it with pre-training, preventing an objective evaluation of the capacities of ChatGPT for DA.

In this paper, we compare the use of ChatGPT for paraphrasing as well as for zero-shot data generation, on five classification datasets (three binary and two multiclass ones), to seven algorithms that have shown good performances in the past. We show that the performance of ChatGPT is highly dependent on the dataset, due mainly to poorly defined tasks, which makes prompting difficult. These tasks were chosen because they are standard on the textual DA literature, and as such these biases are important to point out. With efficient prompting, generating new data with ChatGPT remains the best way to perform textual DA.

## 2 Related Work

There are roughly three main types of DA techniques: word-level augmentation, paraphrase, and generative methods.[1]

In word-level DA, operations modify genuine sentences' words to create variations. Commonly, the operation is word substitution, replacing it with a synonym (Wei and Zou, 2019; Liesting et al., 2021), a neighboring word in pre-trained embedding (Marivate and Sefara, 2020), or by masking and predicting with a neural network (Kobayashi, 2018; Wu et al., 2019; Kumar et al., 2020).

---

[1] For a full review of the literature surrounding textual data augmentation, we refer to (Shorten and Khoshgoftaar, 2019; Feng et al., 2021).

Paraphrasing techniques attempt to create paraphrases from the available sentences. The most seminal technique of this family is Back-Translation (BT), a technique in which a sentence is translated to a pivot language and then back into English (Hayashi et al., 2018; Yu et al., 2018; Edunov et al., 2018; Corbeil and Ghadivel, 2020; AlAwawdeh and Abandah, 2021). Neural networks to directly generate paraphrases have also been used, with specialized decoding techniques for RNN (Kumar et al., 2019), or by using a BART model trained on a corpus of paraphrases generated from BT (Okur et al., 2022).

Generative methods learn the distribution of the training data and generate new data from it. While the obvious advantage is that data should be more diverse, generative models are often more complicated to train and fine-tune correctly. Examples of this family of methods includes using GPT-2 for generating new data (Kumar et al., 2020; Liu et al., 2020; Queiroz Abonizio and Barbon Junior, 2020), other generative models such as VAEs (Malandrakis et al., 2019; Qiu et al., 2020; Piedboeuf and Langlais, 2022) or conditional VAEs to generate examples conditionned on the class (Zhuang et al., 2019; Malandrakis et al., 2019; Rizos et al., 2019; Wang et al., 2020).

Finally, there have also been interest into the use of proprietary models for DA. Both Yoo et al. (2021) and Sahu et al. (2022) show that GPT-3 is able to generate excellent new data, either by completing a list of sentences from one class, or by asking to generate both new sentences and their labels. ChatGPT has also been studied to generate new data, asking it to paraphrase existing data (Dai et al., 2023) for few-shot learning, but the authors first fine-tune the classifier with a large dataset from the same distribution, making it hard to isolate the impact of the generated sentences. Finally, Møller et al. (2023) look at the performance of ChatGPT and GPT-4 for data augmentation compared to human annotation, and conclude that for simple datasets (such as review analysis of products), ChatGPT is better, but otherwise human annotation outperforms data generated by ChatGPT. As they do not compare to other DA techniques, it is also hard to know how ChatGPT performs.

## 3 Algorithms

It is not clear from the literature which DA algorithms perform best, and so in order to thoroughly test the capacities of ChatGPT we select a variety of techniques to compare DA objectively: EDA, AEDA, CBERT, CBART, CGPT, BT, T5-Tapaco, ChatGPT-Par and ChatGPT-Desc. We briefly describe each of those algorithms, and refer to the code for the full details of the implementations and hyper-parameters.[2]

EDA and AEDA are two simple word-level algorithms that achieved great performances in the past. In EDA, one of four operations is chosen (insertion of related words, swapping words, deleting words, and replacing words by synonyms) and applied to a percentage of the words in the sentence.[3] In AEDA (Karimi et al., 2021), punctuations are randomly inserted in the sentence (among "?", ".", ";", ":", "!", and ","), the number of insertion being `RANDINT(1, len(sentence)/3)`.

CBERT and CBART have very similar methodologies. We prepend the class of the example to all genuine sentences, mask a fraction of the tokens, and fine-tune the model on the available training set to predict the masked words. For generation, we then give the modified sentence (masked and with the class prepended) and pass it through the transformers. The main difference between CBERT and CBART is that the latter can predict *spans* instead of tokens, which allows more flexibility.

CGPT also works by prepending the class to the sentence, which then allows GPT-2 to learn to generate conditionally to it. For generation, we give the class as well as the separator token and let GPT-2 generate new sentences. [4]

In BT, we first translate the sentence to a pivot language and retranslate it in English, creating paraphrases. We use the FSMT model from hugging face[5], with the intermediary language being German, which has been shown to obtain good performances (Edunov et al., 2018).

Okur et al. (2022) propose to fine-tune BART on a corpus of in-domain paraphrases created with BT. We found in our experiments that we could get results just as good by using `T5-small-Tapaco`[6], which is the T5 model fine-tuned on the corpus of

---

[2]Code available at https://github.com/smolPixel/DataAugmentationEMNLP2023.

[3]We affect 10% of the words, as recommended in the paper of Wei and Zou (2019)

[4]We use `GPT2-large` with a top_p of 0.95 and a no_repeat_n_gram_size of three.

[5]https://huggingface.co/docs/transformers/model_doc/fsmt

[6]https://huggingface.co/hetpandya/t5-small-tapaco

paraphrases TaPaCo (Scherrer, 2020).

Finally, we test the use ChatGPT, either by asking for paraphrases of genuine sentences (ChatGPT-Par), or by giving a short description of the task and classes and asking for novel sentences (ChatGPT-Desc). We give the exact prompts in Appendix A. Because part of the experiments were done before the API became publicly available, we use in this paper the Pro version of the Web interface of Chat-GPT and leave further fine-tuning for future work[7].

## 4 Datasets and Methodology

We test on five datasets with various complexity and characteristic to fully assess the performance of the algorithms. We use SST-2 (Socher et al., 2013), a binary dataset of movie reviews classification, FakeNews[8], a dataset of news to classify into real news or fake ones, Irony and IronyB (Van Hee et al., 2018), a binary and multiclass version of a task consisting into classifying tweets as ironic or not, and which kind of irony for the multiclass version (polarity clash, situational irony, other irony), and TREC6 (Li and Roth, 2002), a multiclass dataset where the goal is to classify questions into six categories (abbreviation, description, entities, human beings, locations, and numeric values). More information is available in Appendix C.

These datasets were chosen to get a spread of tasks, and because they are commonly used in the literature in data augmentation. SST-2 and TREC6 are both fairly standard in DA research, being used for example in (Kumar et al., 2020; Quteineh et al., 2020; Regina et al., 2021; Kobayashi, 2018). The Irony datasets are also used quite regularly, for example in (Liu et al., 2020; Turban and Kruschwitz, 2022; Yao and Yu, 2021). Finally, while FakeNews has not been used in DA to our knowledge, it is still commonly used for Fake News detection, for example in (Verma et al., 2023; Chakraborty et al., 2023; Iceland, 2023).

We test data augmentation on two settings: few shot learning (10 or 20 starting examples), and classification with dataset sizes of 500 and 1000 examples. While sampling the starting set, we make sure to balance the classes to be able to observe the performance of data augmentation without the additional factor of imbalanced data. We also tested the process on the full dataset but, similarly to other

papers, we found DA to have little effect in this setting. As such we do not discuss them in the paper, but we include them in Appendix C.

Based on results reported in the literature as well as our experiments, we use a larger ratio of generated-to-genuine sentences for small datasets and a smaller one for larger datasets. Concretely, we generate 10 new sentences per genuine sentences for the dataset sizes of 10 and 20, and one for 500 and 1000.[9] As a classifier, we use BERT, and we fine-tune it using the development set, using early-stopping. We run each experiment 15 times, reporting accuracy for binary tasks, and macro-f1 for multiclass tasks.

## 5 Results

Table 1 shows results for the dataset sizes of 10 and 20 and Table 2, for the sizes of 500 and 1000.

For small dataset sizes, we observe that the performance of ChatGPT-Par is on par with the best algorithms, but doesn't beat them by a significant margin on average. While ChatGPT-Desc performs exceptionally well, its performance comes almost exclusively from SST-2, for which it generates highly informative data. For other datasets, it mostly either brings no gain in the performance or a very small one. Overall, all the algorithms provide a reasonable augmentation of the performance, except maybe CGPT, which performs poorly. Excluding ChatGPT-Desc, BART and T5-TaPaCo obtain some of the best performance, although not by much. Given the generative nature of ChatGPT-Desc and its performance, one could also wonder if it would perform better if we generated more sentences. In Appendix C we show that this is not the case, and that the performance for all datasets plateau quickly.

For larger training sets, ChatGPT performs better, while BART and T5-TaPaCo degrade the performance. We believe that this is due to these algorithms creating paraphrases which are closer to the original sentences, leading to less diversity overall. While on few-shot learning this is not a problem because the goal is to give the neural network enough relevant data to learn, on larger datasets diversity seems to become a prevalent factor. Nevertheless, the overall effect of augmentation on moderately

---

[7]This most notably means that we couldn't play with parameters such as the temperature, which affects the diversity of the sentences.

[8]https://www.kaggle.com/c/fake-news/overview

[9]While not a factor much studied in the DA literature, extensive experiments shows that at larger training set sizes, a larger ratio is detrimental since it makes the neural network forget genuine examples.

Table 1: Average metric over 15 runs for the training set sizes of 10 (left) and 20 (right) with a ratio of 10. We report accuracy for binary tasks and macro-f1 for multiclass ones. STDs are between 1.5 and 5, depending on the dataset.

| | SST2 | FakeNews | Irony | IronyB | Trec6 | Average |
|---|---|---|---|---|---|---|
| Baseline | 56.4/60.6 | 52.2/53.3 | 52.5/58.3 | 23.6/23.4 | 27.9/34.7 | 42.5/46.1 |
| EDA | 59.4/63.2 | 55.0/56.6 | 53.7/57.3 | 25.3/27.8 | 30.8/43.7 | 44.9/49.7 |
| AEDA | 59.3/64.6 | 53.4/55.3 | 54.1/56.0 | 25.4/27.1 | 26.5/44.2 | 43.7/49.5 |
| BT | 59.0/64.6 | 55.1/56.2 | 54.3/56.5 | 25.4/26.7 | 32.9/**46.3** | 45.4/50.1 |
| CBERT | 57.6/63.1 | 54.5/55.5 | **55.1**/57.4 | 25.3/29.5 | 29.0/40.6 | 44.3/49.2 |
| CGPT | 55.6/61.2 | 52.4/54.7 | 51.9/53.1 | 22.9/25.1 | 23.3/38.1 | 41.2/46.4 |
| CBART | 60.5/64.9 | **55.8/57.2** | 54.8/56.6 | 25.5/**28.0** | **34.1**/46.1 | 46.2/50.6 |
| T5 | 60.4/64.6 | 54.5/56.6 | 53.2/56.7 | 23.5/27.1 | 34.0/**46.3** | 45.1/50.3 |
| GPT3.5-Par | 62.5/69.0 | 53.8/54.9 | 54.2/**57.5** | 24.4/27.8 | 31.3/44.8 | 45.3/50.8 |
| GPT3.5-Desc | **78.6/82.6** | 51.5/52.8 | 53.2/54.1 | **27.1**/27.7 | 31.4/42.9 | **48.3/52.0** |

Table 2: Average metric over 15 runs for the training set sizes of 500 (left) and 1000 (right) with a ratio of 1. We report accuracy for binary tasks and macro-f1 for multiclass ones. STDs are between 0.6 and 3.0, depending on the dataset.

| | SST2 | FakeNews | Irony | IronyB | Trec6 | Average |
|---|---|---|---|---|---|---|
| Baseline | 87.7/88.8 | 73.3/77.0 | 65.6/68.1 | 42.4/45.2 | 81.0/85.4 | 70.0/72.9 |
| EDA | 87.9/88.9 | 73.7/77.6 | 65.8/68.8 | **43.1/46.6** | 81.3/86.1 | 70.4/73.6 |
| AEDA | 88.0/89.0 | 73.5/77.6 | 65.7/69.2 | 42.8/46.1 | **82.7**/86.4 | 70.5/73.7 |
| BT | **88.2/89.1** | 73.6/77.4 | 66.2/69.2 | 42.4/46.0 | 81.7/86.1 | 70.4/73.5 |
| CBERT | 87.5/88.3 | 73.6/77.5 | 65.8/68.1 | 40.6/45.5 | 80.9/85.4 | 69.7/72.9 |
| CGPT | 87.8/88.7 | 73.2/77.6 | 65.2/68.8 | 42.8/45.2 | 82.1/**87.2** | 70.2/73.5 |
| CBART | 87.7/88.6 | **73.9/77.9** | 65.9/68.8 | 42.7/45.0 | 78.6/83.4 | 69.8/72.7 |
| T5 | 87.9/88.7 | 73.8/73.3 | 65.3/68.2 | **43.1**/45.3 | 79.9/85.0 | 70.0/72.1 |
| GPT3.5-Par | **88.2/89.1** | 73.8/77.7 | **66.8/69.3** | 42.8/45.9 | 82.4/87.1 | **70.8/73.8** |
| GPT3.5-Desc | 87.4/88.9 | 71.9/75.9 | 64.1/66.9 | 41.1/44.4 | 79.8/84.0 | 68.9/72.0 |

sized datasets are very small, which brings the question of whether data augmentation is relevant at all in these cases.

# 6 Discussion

Of all the algorithms used here, T5 and ChatGPT present the greatest novelty as well as show some of the best performances. As such, we center our discussion on these two algorithms. When we observe T5 sentences (see Appendix B), we can see that they are not as good as one would expect, often being ungrammatical or badly formed. Still, it has been noted before that having correctly formed sentences is not an important criterion for the efficiency of a DA algorithm, which might explain why its performance is high (Karimi et al., 2021).

ChatGPT-Desc often has difficulty generating sentences of the desired class, accounting for its overall poor performance, while ChatGPT-Par creates excellent paraphrases, bringing diversity to the dataset while maintaining class coherence. Nevertheless, there is a hidden cost that we found was not

discussed in other papers, namely the need for data reparation. ChatGPT-Par quite often refuses to create paraphrases, especially for the FakeNews and Irony/IronyB datasets which contain occasional mentions of *rape* and *sex*. In these cases, we had to manually find the "bad" element of the batch and rerun it without it, adding considerable complexity to the data augmentation algorithm. Another option would be to simply not correct them, but our preliminary studies indicate that this degrades the performance of DA significantly.[10]

## 6.1 Poorly defined tasks and dataset biases.

Despite the description strategy being able to add something akin to external data, our experiments show that ChatGPT underperforms with this method, the performance often being worse than when paraphrasing the existing data. This raises many questions as adding more diverse data should augment performance.

[10]As we see in Appendix C, running with a smaller ratio for few shot learning reduces the performance by a large amount.

We found that for most part, the poor performance of ChatGPT was related to the poor health of the datasets. Except for SST-2, we found that FakeNews, Irony, IronyB, and TREC6 have poorly defined labels in relation to the task, and that examples in the datasets were often ambiguous to human eyes. Under these conditions, it is difficult to expect ChatGPT to perform well. We underline these problems here because poor dataset health is not a rare phenomenon.

Irony and IronyB are two datasets of the SemEVAL 2018 competition. Data was collected from Twitter by collecting tweets that contained some specific hashtag such as #irony, #not, or #sarcasm, which were then removed to form the data of the ironic class. The non-ironic class was then formed by collecting other tweets. This creates a heavy bias in the dataset which shift the task from predicting if the tweet is ironic to predicting if there was a #irony hashtag coming with it. Without the clue that the hashtags give us, it is often impossible to know if the tweet is ironic or not, and we show in Appendix C some examples of ironic tweets which, from manual labelling, we found were ambiguous in their classes.

TREC6 is a dataset of the Text REtrieval Conference and consists in classifying the questions into six categories. While all data was manually annotated, we found inconsistencies in the annotation. For example, "What is narcolepsy?" is labelled as *description* but "What is a fear of motion?" as an *Entity*. Other inconsistencies are "What is the oldest profession?" and "What team did baseball 's St. Louis Browns become?" labelled as Human vs "What do you call a professional map drawer" as Entity, or "Where did Indian Pudding come from?" being labelled as Description but "Where does chocolate come from?" as Location. Given that the same mislabelling remains in the test set (ex "Where does dew come from?" being labelled as location), ChatGPT generating sentences of the correct class won't help the classifier much. It is to note that these issues were already noted by Li and Roth (2002), who advise using multi-label classification to reduce biases introduced in the classifier. In all of its usage for DA however, we found it used as a regular classification problem, with all the ambiguity problems it entails.

Finally, FakeNews is a Kaggle dataset which has been thereafter used in many papers for fake news detection. We decided to use this dataset because

it seemed a difficult and interesting task, but while analyzing it, we found it biased in a sense similar to Irony and IronyB. From what little information we could find, news were gathered from various sources and split into real or fake news based on the journal they came from. This causes biases because while some journals may have a tendency to sprout fake news, it does not mean all of its news are fake. Furthermore, we found strange choices of labelling. For example, all articles from Breitbart are labelled as real news even if it receives a mixed score of factual reporting[11] and articles from consortium news, which receives the overall same score, are labelled as fake[12].

By refining prompting, we can augment the TREC6 dataset to go to 68.6, which still underperforms when compared to BERT[13]. We found ChatGPT to have difficulty understanding the concept of "Entity" and "Human" questions, often labelling them instead as "Description".

## 7 Conclusion

Data augmentation is a seminal technique to lower annotation cost and keep good performances, but even today it is difficult to figure out which technique works best. In particular, the use of ChatGPT has not been correctly assessed for data augmentation, leading to the unknown factor for industries of whether it is worth the price.

In this paper, we study nine data augmentation techniques, including a novel one using a pretrained T5 system for paraphrasing, and show that while ChatGPT achieves among the best results, it doesn't outperform the other algorithm by a significant margin. This, coupled with the fact that using ChatGPT costs both time and money when compared to the other algorithms, brings us to a different conclusion than what previous studies using ChatGPT for DA found, namely that it might not be worth it depending on the task. We further found that while zero-shot generation of data could give outstanding results, it was often hindered by biased datasets, which prevented efficient prompting of ChatGPT.

---

[11]https://mediabiasfactcheck.com/breitbart/
[12]https://mediabiasfactcheck.com/consortium-news/
[13]Because FakeNews and the Irony datasets contains a lot of examples breaking ChatGPT term of services, we could not attempt to refine the prompting for those datasets

## Limitations

This paper explores the use of ChatGPT for DA, comparing it to other algorithms from the literature. Technical limitations of this paper include limited fine-tuning for some of the algorithms, including ChatGPT for which we used the Web interface and therefore could not finetune the hyperparameters. While the other algorithms have been fine-tuned on some hyper-parameters, fine-tuning was built on some supposition (such as the use of German as a pivot language for BT), which may not be the best.

This paper also focuses on English language and textual classification for short sentences, both assumptions which do not hold for many other tasks. As such, we do not guarantee the results are applicable to other tasks, especially for languages which are low-resource (such as Inuktitut or Swahili) or for longer texts, for which most of the algorithms used would most likely perform poorly due to lack of training data/limited context for input.

## Ethics Statement

Use of pre-trained language models, and especially of "very large language models", come with a plethora of ethical problems which have been well discussed in the litterature, including the environmental cost (Schwartz et al., 2019; Bender et al., 2021) and environmental racism (Rillig et al., 2023), the repetition of learned biases against minorities (Singh, 2023), and concerns over data privacy (Li et al., 2023).

A big concern with the most recent models is the effect it will have on employement, but we believe this paper mitigates this effect by showing limitation of ChatGPT, especially in the context of data annotation and dataset creation.

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

## A  Prompts used for ChatGPT

We use three types of prompts for querying ChatGPT, two for the paraphrasing technique and one for the description technique, in order to get the best possible results while minimizing the number of queries. For the paraphrasing, if the ratio was more than one, then the query is of the type: "Create X paraphrases of the following sentence : "

If the ratio is one, then we process the examples in batch, with the query: " Create a paraphrase for each of the following sentences: 1. [...], 2. [...]"

Finally, for the description strategy the template we use is : "Generate 10 new sentences that you haven't generated before for a dataset of DATASET_DESCRIPTION which would be CLASS_DESCRIPTION". We found that specifying "new sentences that you haven't generated before" helped ChatGPT generate more diverse sentences. The given dataset descriptions are "movie review", "headline Fake/Real news classification", "Ironic tweet detection", and "Question Classification".

The class values are "negative or somewhat negative" or "positive or somewhat positive" for SST-2, "Real" and "Fake" for FakeNews, "Non Ironic Tweets" and "Ironic Tweets" for Irony, "Tweets ironic by polarity contrast, where the polarity is inverted between the literal and intended evaluation", "Tweets ironic by Situational Irony, where a situation fails to meet some expectation", "Tweets ironic by Other type of Irony, where the Irony is neither by Polarity Contrast or by Situational Irony", and "Tweets that are not ironic" for IronyB, and finally for TREC6 we use "Question about an abbreviation" , "Question about an entity (event, animal, language, etc)", "Question concerning a description (of something, a definition, a reason, etc)", "Question about a human (description of someone, an individual, etc)", "Question about a location", and "Question about something numerical (weight, price, any other number)".

We referred to the description given in the original papers of each dataset to craft informative prompts.

## B  Example of generated sentences

We give in Table 3 examples of generated sentences for the SST-2 dataset and the negative class, with the starting sentence "makes a joke out of car chases for an hour and then gives us half an hour of car chases." for the algorithms that takes a sentence as an input (all except CGPT and ChatGPT-Desc). When fine-tuning is needed, we use a training set size of 20.

| Algo | Generated sentence |
|------|-------------------|
| EDA | makes a joke out as of car chases for an hour and then gives us half an hour of car chases . |
| AEDA | makes , a joke out of? car chases for an ! hour and , then gives us ! half an hour of : car , chases . |
| BT | turns car chases into a joke for an hour and then gives us half an hour of car chases. |
| CBERT | makes two breakfast out of car wash for a,, then gives us half an inch of car wash. |
| CGPT | 'makes a joke out of car chases for an hour and then gives us half an |
| CBART | Stays out of car chases for an hour and then gives up on an hour's worth'n'a-bit-of-car-chases |
| T5-Tapaco | The car chases for half hour is a joke. |
| ChatGPT-Par | It turns car chases into a comedic spectacle for an entire hour, followed by another 30 minutes of non-stop car action. |
| ChatGPT-Desc | The film was a major disappointment, lacking any coherent plot or engaging characters. |

Table 3: Examples of generated sentences for each algorithm for the SST-2 dataset and with a dataset size of 20.

## C  Supplementary Results

In this section we give supplementary results to the paper. Table 4 gives some information about the datasets, and Table 5, the results of data augmentation on the full training set, with a ratio of generated-to-genuine of one.

Table 4: The tasks tackled in this study. The length of the sentences is defined by the number of tokens when tokenized at white spaces.

| Name | SST2 | Irony | FakeNews | IronyB | TREC6 |
|------|------|-------|----------|--------|-------|
| \|classes\| | 2 | 2 | 2 | 4 | 6 |
| $\|D_{\text{train}}\|$ | 6920 | 2683 | 12799 | 2681 | 5452 |
| len. sents. | 19.3 | 13.7 | 12.5 | 13.7 | 10.2 |

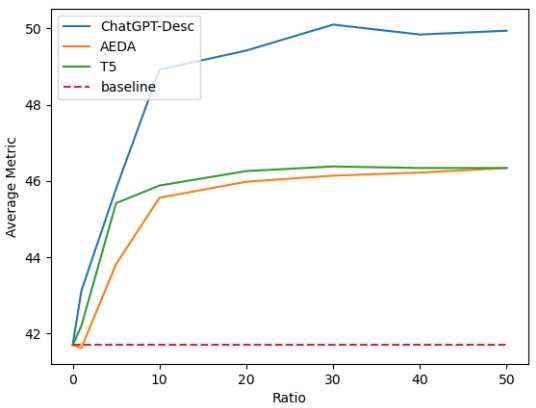

Figure 1: Average metric vs Ratio for a dataset size of 10 and ChatGPT-Desc, AEDA, and T5.

Figure 1 shows the performance as we increase the ratio for the ChatGPT-Desc strategy, compared to AEDA and T5, and with a dataset size of 10. As we can observe, the performance plateau quickly for all algorithms. Given that Chat-GPTDesc performs much better on SST-2 than the other datasets, we also give in Figure 2 the results while excluding SST-2. We leave for future work to investigate whether the plateauing for ChatGPT-Desc is due to lack of fine-tuning or simply the limit of ChatGPT when it comes to generate diverse sentences.

## D  Technical details

All final hyperparameters are detailed in the github, and we show a summary of which hyperparameters we fine-tuned in Table 7. For fine-tuning the classifiers, we changed the number of epochs while leaving the other parameters fixed. For fine-tuning the algorithms, we played with the hyperparameters detailed in Table 7, exploring random combinations around the hyperparameters recommended in the original papers. To correctly assess the capacities of the DA methods on the different datasets, we keep the same hyperparameters for a given dataset size across all datasets. Experiments were run on NVIDIA GeForce RTX 3090 with 24G of memory.

Table 5: Average metric over 15 runs (accuracy and macro-F1) for the full training set and for all datasets. STDs are between 0.3 and 3.3, depending on the dataset.

|            | SST2 | FakeNews | Irony | IronyB | Trec6 | Average |
|------------|------|----------|-------|--------|-------|---------|
| Baseline   | 87.7 | 73.3     | 65.6  | 42.4   | 81.0  | 70.0    |
| EDA        | 87.9 | 73.7     | 65.8  | **43.1** | 81.3 | 70.4    |
| AEDA       | 88.0 | 73.5     | 65.7  | 42.8   | **82.7** | 70.5  |
| BT         | **88.2** | 73.6 | 66.2  | 42.4   | 81.7  | 70.4    |
| CBERT      | 87.5 | 73.6     | 65.8  | 40.6   | 80.9  | 69.7    |
| CGPT       | 87.8 | 73.2     | 65.2  | 42.8   | 82.1  | 70.2    |
| CBART      | 87.7 | **73.9** | 65.9  | 42.7   | 78.6  | 69.8    |
| T5         | 87.9 | 73.8     | 65.3  | **43.1** | 79.9 | 70.0    |
| GPT3.5-Par | **88.2** | 73.8 | **66.8** | 42.8 | 82.4 | **70.8** |
| GPT3.5-Desc| 87.4 | 71.9     | 64.1  | 41.1   | 79.8  | 68.9    |

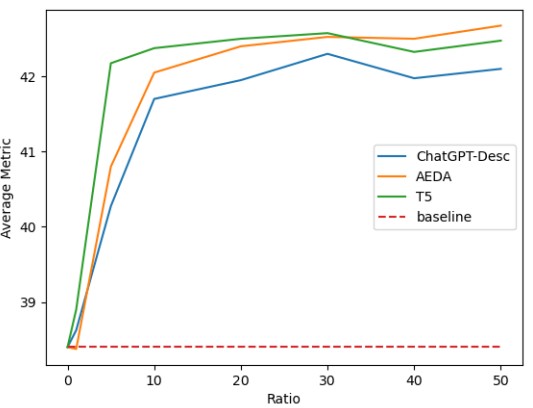

Figure 2: Average metric vs Ratio for a dataset size of 10 and ChatGPT-Desc, AEDA, and T5, excluding the SST-2 datasets from the average.

| EDA    | ratios ins, del, swap, syn |
|--------|----------------------------|
| AEDA   | -                          |
| BT     | num. beams (en $\rightarrow$ de, de $\rightarrow$ en), |
| CBERT  | lr, epochs, bs, ratio mask |
| CGPT   | lr, epochs, bs,            |
| CBART  | lr, epochs, bs, ratio mask |
| T5     | top_p                      |
| ChatGPT| -                          |

Table 7: Hyperparameters fine-tuned during our experiments. lr stands for learning rate, bs for batch size, and ratio mask refers to the percentage of words that are masked.

|        | 10    | 20    | 500   | 1000  |
|--------|-------|-------|-------|-------|
| EDA    | 0m1s  | 0m1s  | 0m4s  | 0m6s  |
| AEDA   | 0m0s  | 0m0s  | 0m0s  | 0m0s  |
| BT     | 0m7s  | 0m11s | 0m27s | 0m52s |
| CBERT  | 0m1s  | 0m1s  | 0m8s  | 0m15s |
| CGPT   | 0m15s | 0m22s | 2m11s | 5m38s |
| T5     | 0m6s  | 0m3s  | 0m18s | 0m36s |

Table 8: Running time for the different DA algorithms and SST-2, excluding the time of the classifier.

In Table 8, we show the running time for augmentation part of the process, for SST-2 and the different dataset sizes. We do not include the training time of the classifier.

| Ironic     | Shoutout to my mom for being hella supportive of me |
|------------|------------------------------------------------------|
| Ironic     | Luv this                                             |
| Non-Ironic | @alyssaanicoleL this Friday lit                      |
| Non-Ironic | they don't sing live, but they sure are hella good looking #smh |

Table 6: Examples of the Irony datasets which our manual examination found to be ambiguous.