# OpenReview forum: "Is ChatGPT the ultimate Data Augmentation Algorithm?"
_EMNLP/2023/Conference — EMNLP 2023 Findings_

### Official Review · Reviewer_Fjbw · 2023-08-03

**Soundness:** 3

**Excitement:**

2: Mediocre: This paper makes marginal contributions (vs non-contemporaneous work), so I would rather not see it in the conference.

**Missing References:**

1. Line 140, there should be references to papers about EDA, CBERT, CBART, and CGPT.


**Paper Topic And Main Contributions:**

This paper evaluated the potential of applying ChatGPT for data augmentation, which can be essential for dataset construction and improving model performance. This paper conducted experiments on five classification tasks by comparing data augmentation by ChatGPT for paraphrasing and zero-shot data generation with seven baselines. The paper identified several limitations of applying ChatGPT for DA such as difficulty generating sentences of the desired class and the need for data reparation.

**Reasons To Accept:**

1. This paper explored ChatGPT's performance on DA, which has not been well investigated.
2. The experiments setting is reasonable and sufficient.
3. The paper identified limitations of both applying DA for text classification tasks, as well as limitations of applying ChatGPT for DA, which can be useful for future research.

**Reasons To Reject:**

1. The paper is generally an evaluation of ChatGPT's ability, which makes the contribution limited.
2. The analysis in this paper is rather shallow. We encourage the authors to conduct detailed and in-depth analysis on the augmentated examples, and support the conclusion with more visible and stronger evidence.


**Reproducibility:**

3: Could reproduce the results with some difficulty. The settings of parameters are underspecified or subjectively determined; the training/evaluation data are not widely available.

**Reviewer Confidence:**

3: Pretty sure, but there's a chance I missed something. Although I have a good feel for this area in general, I did not carefully check the paper's details, e.g., the math, experimental design, or novelty.

**Typos Grammar Style And Presentation Improvements:**

Grammar issue:
1. In the Section Algorithms, there should be some brief introduction to the baseline methods and acronyms, which would make it easier to follow.
2. Table 5 reports the hyperparameters fine-tuned during experiments, however, the value of these paprameters are not reported.

---

> ### Author Rebuttal · Authors · 2023-08-24
>
> Thank you very much for your comments. We agree that the contribution is limited to the ability of ChatGPT, but we would like to argue that it is an essential contribution in a time when many papers are claiming the superiority of ChatGPT in data creation/curating tasks. Showing that it possesses very real limits and that it does not fair well above others, carefully constructed, and most importantly, free algorithms is we believe an interesting contribution, at least for a short paper.
>
> We fully agree on the need of more analysis, and we will provide more in depth analysis if the paper is accepted.
>
> For the citations missing on line 140, we cited all relevant material in the related work section, but have no problem to duplicate the citation. Thanks for suggesting it. We will as well add an introduction to the baseline method and acronyms.
>
> All hyperparameters (as well as the data we used) are available in the supplementary zip file, which contains the code for running the experiments. But definitely, we will add hyperparameters in the appendix.

---

### Official Review · Reviewer_WQnu · 2023-08-03

**Soundness:** 2

**Excitement:**

3: Ambivalent: It has merits (e.g., it reports state-of-the-art results, the idea is nice), but there are key weaknesses (e.g., it describes incremental work), and it can significantly benefit from another round of revision. However, I won't object to accepting it if my co-reviewers champion it.

**Paper Topic And Main Contributions:**

The author discussed the potential of using GPT for text augmentation and concluded that while GPT can lead to some improvement in existing tasks, the overall enhancement is not significant. Moreover, the high cost of training GPT makes it less suitable for text augmentation tasks.

**Questions For The Authors:**

Please refer to Reasons To Reject.

**Reasons To Accept:**

The author's experiments may indeed provide valuable insights and inspiration to the GPT community. They highlight the potential of using GPT for text augmentation and demonstrate its impact on existing tasks. While the results may not show significant improvements in isolation, they can spur further research and encourage exploration of innovative ways to combine GPT with other techniques to enhance various natural language processing tasks.






**Reasons To Reject:**

The author's conclusion may be one-sided. The data obtained by using GPT to enhance prediction should be integrated with existing methods to achieve better results, such as combining with comparative learning.



**Reproducibility:**

2: Would be hard pressed to reproduce the results. The contribution depends on data that are simply not available outside the author's institution or consortium; not enough details are provided.

**Reviewer Confidence:**

2: Willing to defend my evaluation, but it is fairly likely that I missed some details, didn't understand some central points, or can't be sure about the novelty of the work.

---

> ### Author Rebuttal · Authors · 2023-08-24
>
> Thank you very much for your comments. We agree that we may get better performances by combining DA with another form of learning. However, we feel this falls outside the scope of the paper where we aimed to objectively assess the capacity of ChatGPT for DA, as DA alone is something that is still heavily researched.
>
> Regarding «Soundness»: We do believe working on DA alone is worth the investigation, but this is of course arguable.
>
> Regarding «Reproducibility»: We understand your concern: this is a short paper, and since we compare many techniques, it is hard to provide a self-contained description of all of them. Still, we work on public datasets, and we provide a documented code necessary to reproduce all the results we produced in the supplementary materials (code, hyperparameters, and data). We propose to add a complete description of each method in the appendix.

---

### Official Review · Reviewer_FG9i · 2023-08-05

**Soundness:** 3

**Excitement:**

3: Ambivalent: It has merits (e.g., it reports state-of-the-art results, the idea is nice), but there are key weaknesses (e.g., it describes incremental work), and it can significantly benefit from another round of revision. However, I won't object to accepting it if my co-reviewers champion it.

**Missing References:**

Some works refer to GPT augmentation as weak supervision. Some discussion there would be helpful.

**Paper Topic And Main Contributions:**

The author compared the ability of ChatGPT in paraphrasing and data augmentation and tested the performance on 5 classification datasets with 7 other augmentation algorithms. The results showed that GPT augmentation can be limited when it comes to sensitive topics, and overall it is not the best augmentation method even though it has good performance on simpler datasets.

**Reasons To Accept:**

The author selected representative groups of existing DA methods and provided a reasonable amount of experiments. Also, the paper is well-structured.

**Reasons To Reject:**

1. I recommend some brief explanations for how the selected algorithm works, such as CBERT, CBART, and CGPT since they are not discussed in the related work and no detailed explanation in the algorithm part.
2. In the related work, I suggest discussing how's the performance of the DA by GPT or other methods.
3. I don't see the point to generate 1 augmentation for the larger training size. I would suggest keeping the same ratio between augmented data and original data. Based on the results, there's nothing much the 1 augmented example can do and all differences may come from the randomness.

**Reproducibility:**

4: Could mostly reproduce the results, but there may be some variation because of sample variance or minor variations in their interpretation of the protocol or method.

**Reviewer Confidence:**

5: Positive that my evaluation is correct. I read the paper very carefully and I am very familiar with related work.

---

> ### Author Rebuttal · Authors · 2023-08-24
>
> Thank you very much for your comments.
>
> Regarding reason n.1 for Rejection, section 3 is dedicated to this effect (line 128 to 183, 148-162 for CBERT, CBART, and CGPT). Short formats only allow for general discussion, but we will review our descriptions to make them clearer.
>
> Rejection reason n.2: we cited work using ChatGPT for data augmentation (line 37, line 114-127), to our knowledge the only two existing at the time of writing, and explained their conclusions. If we are missing any important references that we didn't see during our literature review, please do not hesitate to point them out.
>
> Rejection reason n.3:   We attempted many ratios of generated-to-genuine, and found that for larger seed size, there were no benefits to have larger ratios. This is because DA is generally ineffective on larger dataset size except for a weak regularization effect, which can be attained with few data (compared with few shot learning where DA acts to prevent overfitting). This inefficiency has been well documented in the literature. We agree however that it can be misleading and will better explain our reasoning.

---

### Meta-Review · Area_Chair_kdTT · 2023-09-19

**Recommendation:** 3

**Metareview:**

The authors used ChatGPT to create new data both with paraphrasing and with zero-shot generation, and compare it to seven other algorithms.
The 2 out of 3 reviewers selected "Good" for soundness and reproducibility seem to have a problem.

---

### Decision · Program_Chairs · 2023-10-07

**Decision:**

Accept-Findings

**Comment:**

The authors used ChatGPT to create new data both with paraphrasing and with zero-shot generation, and compare it to seven other algorithms.
The 2 out of 3 reviewers selected "Good" for soundness and reproducibility seem to have a problem.